# Proposal for a continuous wave laser with linewidth well below the standard quantum limit

Chenxu Liu [1,2,3 ✉], Maria Mucci[1,2], Xi Cao[1,2], M. V. Gurudev Dutt[1,2], Michael Hatridge[1,2] & David Pekker [1,2 ✉]

Due to their high coherence, lasers are ubiquitous tools in science. We show that by engineering the coupling between the gain medium and the laser cavity as well as the laser cavity and the output port, it is possible to eliminate most of the noise due to photons entering as well as leaving the laser cavity. Hence, it is possible to reduce the laser linewidth by a factor equal to the number of photons in the laser cavity below the standard quantum limit. We design and theoretically analyze a superconducting circuit that uses Josephson junctions, capacitors and inductors to implement a microwave laser, including the low-noise couplers that allow the design to surpass the standard quantum limit. Our proposal relies on the elements of superconducting quantum information, and thus is an example of how quantum engineering techniques can inspire us to re-imagine the limits of conventional quantum systems.

[1] Department of Physics and Astronomy, University of Pittsburgh, Pittsburgh, PA 15260, USA. [2] Pittsburgh Quantum Institute, University of Pittsburgh, Pittsburgh, PA 15260, USA. [3] Present address: Department of Physics, Virginia Tech, Blacksburg, VA 24061, USA. ✉email: chenxu_liu@pitt.edu; pekkerd@pitt.edu

The key property of a laser, which is crucial to its applications in quantum optics, quantum information, and metrology, is its high coherence or narrow linewidth. The main components of a laser, depicted in Fig. 1, are (1) one or more atoms with an inverted population (also called the gain medium), (2) an atom-cavity coupler, (3) a lasing cavity, and (4) a cavity-output coupler and an output channel (transmission line). The standard quantum limit (SQL) for the phase coherence time was first introduced by Schawlow and Townes[1], who showed that the minimum possible laser linewidth is determined by the linewidth of the laser cavity divided by twice the number of photons in the cavity. This raises the question whether it is possible to surpass this limit? Previous work on laser theory[2], quantum-cascade lasers[3], superradiant lasers[4], and number-squeezed lasers[5] has focused on the gain medium (1) and the atom-cavity coupler (2). Specifically, in Ref. [2], Wiseman showed theoretically that by using Susskind–Glogower operators[6] to couple the gain medium to the laser cavity, it is possible to eliminate pump noise, but not loss noise[7]. This decreases the minimum laser linewidth, though only by a factor of two.

At optical frequencies, the inherent light-matter coupling is rather weak and consequently optical devices tend to be only weakly nonlinear. Over the past two decades, significant progress has been made on building strongly nonlinear optics. At optical frequencies, using small-size high-Q optical cavities coupling to atoms, strong coupling can be achieved in cavity QED system by reducing the optical mode volume[8–11]. On the other hand, at microwave frequencies, circuit QED achieves extremely strong light-artificial atom interactions by utilizing the extreme non-linearity and small size (compared to microwave-frequency photons) of Josephson junctions[12]. Circuit QED devices include the various flavors of superconducting quantum computing platforms with components like fluxonium[13] and transmon qubits[14,15], resonant cavities, microwave waveguides, and quantum limited parametric amplifiers[16,17]. There is also experimental precedent for building conventional lasers using superconducting circuits with linear couplers[18–20], as well as devices based on parametrically driven, weakly nonlinear oscillators[21,22].

In a very recent work, Baker et al.[23] pointed out that it should be possible to reduce the linewidth of a laser by a factor of $\sim\langle n\rangle^2$ below the SQL, where $\langle n\rangle$ is the mean photon number inside the cavity. They called this new limit on laser linewidth the "Heisenberg limit." Furthermore, Baker et al. proposed a microwave

circuit in which photon gain and loss processes were engineered using a pair of "photon treadmills" to add and remove photons from the microwave laser (maser) cavity. Baker's photon treadmill proposal represents a substantial increase in complexity, as this maser requires multiple, pulsed light sources that make the maser's linewidth a direct tradeoff against complex controls.

In the present work, we begin by showing that simple engineering of both the cavity-output coupling, in addition to the atom-cavity coupling (see Fig. 1), can be used to suppress the phase noise in the lasing cavity. Although our simple engineering results in a decrease of the laser linewidth by a factor $\sim\langle n\rangle$ below the SQL, as compared with $\sim\langle n\rangle^2$ obtained by Baker et al., it provides us with useful notions for how to construct lasers that operate well below the SQL using only static couplers that do not require coherent light pulses. Using these notions, we establish our main result: a blueprint for building a maser in which Josephson junctions and linear inductors are used to construct nonlinear coupling circuits. These coupling circuits approximate the behavior of Susskind–Glogower operators for a range of cavity photon occupancies. We show that by using these circuits to couple the laser cavity to both the gain medium and the output port, it is possible to suppress the linewidth of the resulting maser beyond the SQL by a factor of $\langle n\rangle^{-1.098}$. Although we do not achieve the proposed "Heisenberg limit" of $\langle n\rangle^2$ reduction, our scheme requires only a single, continuous, incoherent pump and is thus a much simpler light source to control. We believe that this is an important advantage for the development of sub-SQL lasers at optical frequencies as well as for applications of these devices both at microwave and optical frequencies.

## Result

**Overview of the results.** The starting point for our exploration is a proposal due to Wiseman[2] for reducing laser spectral linewidth. The laser linewidth, coming from the phase noise of the laser light, has two equal contributions from both the atom-pump process and the cavity loss process[7]. In ref. [2], Wiseman proposed using "bare" (so called because they lack a photon number scaling pre-factor) raising and lowering operators

$$\hat{e} = \sum_i |i\rangle\langle i+1| \quad \hat{e}^\dagger = \sum_i |i+1\rangle\langle i|, \qquad (1)$$

which were first introduced by Susskind and Glogower[6], to couple atoms of the gain medium to the resonant cavity of the laser. As

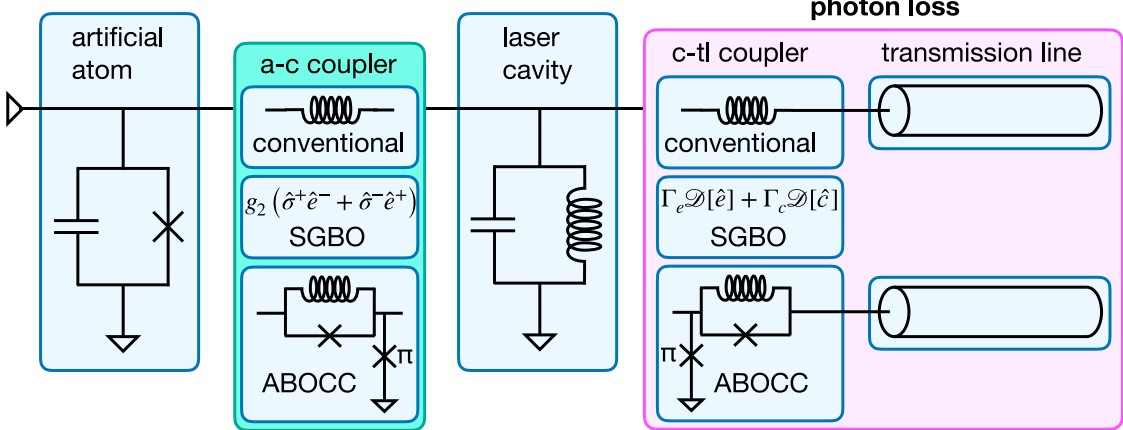

**Fig. 1 Schematic of the superconducting Josephson laser.** The laser is composed of: an artificial two-level atom (transmon qubit) that is incoherently pumped from the ground to the excited state at the rate $\Gamma_p$, an atom-cavity (a-c) coupling circuit, a laser cavity made of an LC resonator, a cavity-transmission line (c-tl) coupling circuit, and an output transmission line. The coupling circuits come in three flavors: (1) conventional: linear inductors, (2) Susskind–Glogower bare operator (SGBO) couplers, and (3) Approximate Bare Operator Coupling Circuits (ABOCC). As the SGBO scheme is a purely theoretical construct, it is represented by the a-c coupling Hamiltonian and the photon loss operator.

these operators commute with the phase $\hat{\phi}$ of the optical field in the cavity (which can be verified by observing that $\hat{e} = e^{i\hat{\phi}}$), Wiseman's proposal eliminates pump noise and therefore reduces the minimum linewidth by a factor of two (labeled $D_{ST2}$) below the SQL, also known as the Schawlow–Townes (ST) limit (labeled $D_{ST}$)[2,24], i.e.,

$$D_{ST2} = \frac{1}{2}D_{ST}, \quad D_{ST} = \frac{\Gamma_c}{4\langle n \rangle}, \tag{2}$$

where $\Gamma_c$ is the cavity linewidth and $\langle n \rangle$ is the mean photon number inside the cavity.

Our first result is a simple argument that establishes the result that was presented by Baker et al.[23], that it is possible to surpass the SQL on laser linewidth by manipulating both that the atom-cavity and the cavity-output channel couplings. We show that the application of Wiseman's scheme to both couplings can be used to eliminate both the pump and the cavity loss noise. However, we must also add some conventional loss to the laser in order to stabilize it, the result is a linewidth that is $\langle n \rangle$ times narrower than the ST limit. Having a mathematical scheme for building an ultranarrow linewidth laser, we still need an experimentally viable method for building bare operators.

Our second, and main, result is an Approximate Bare Operator Coupling Circuit (ABOCC), composed of Josephson junctions and inductors, that approximates the desired coupling Hamiltonian over a range of cavity photon occupancies. For laser applications, our ABOCC is an attractive alternative to previous proposals to build bare operators that relied on adiabatic rapid passage[2,25–29] as it does not require additional drives. We argue that an ABOCC laser, in which couplers are ABOCCs, is a practical laser design that achieves the ultranarrow linewidth promised by Susskind-Glogower bare operator (SGBO) laser (see Fig. 1). In the remainder of this paper, we first calculate the behavior of the purely theoretical SGBO laser compared to an ideal conventional laser, and then describe the physically realizable ABOCC in detail and describe the potential performance of a laser based on pair of ABOCCs.

In order to compare the linewidth $D$ of different laser designs to the ST limit $D_{ST}$ we need to generalize the ST formula for the cases in which the cavity-transmission line coupler is not linear. We do so by replacing the cavity linewidth $\Gamma_c$ by the ratio of the laser luminosity to the energy of the photons in the cavity $\Gamma_c \to P_{out}/(\hbar\omega_c\langle n \rangle)$ in Eq. (2) thus obtaining the formula

$$D_{ST} = \frac{P_{out}}{4\hbar\omega_c\langle n \rangle^2}. \tag{3}$$

For conventional lasers, Eq. (3) is identical to the standard ST linewidth formula. To ensure that each type of laser is performing at its optimal, we fix the mean photon number in the cavity and minimize the ratio $D/D_{ST}$ by tuning the laser parameters. For example, for the case of the conventional laser, we tune the atom-cavity coupling strength ratio $g/\Gamma_c$ and the atom incoherent pump rate ratio $\Gamma_p/\Gamma_c$.

In Fig. 2 we plot the optimum laser linewidth, relative to the ST limit, as a function of the average number of photons in the laser cavity for three types of lasers: the conventional laser, the SGBO laser, and the ABOCC laser. All data in this figure were obtained numerically using the spectral method to analyze the master equation (see Methods). We observe that for the conventional laser, the ratio $D/D_{ST}$ approaches unity as $n$ becomes large. At the same time, we observe that the laser linewidth for the SGBO laser as well as the ABOCC laser is significantly narrower and goes as $D \sim D_{ST}/\langle n \rangle$.

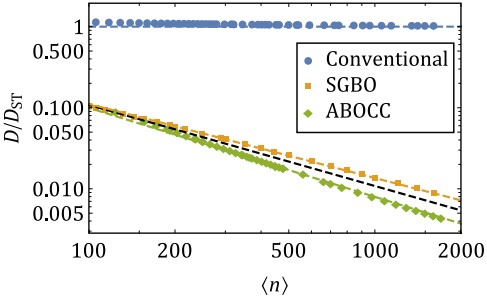

**Fig. 2 Laser linewidth as a function of the average number of photons ⟨n⟩ in the laser cavity for conventional, Susskind–Glogower Bare Operator (SGBO), and Approximately Bare Operator Coupling Circuit (ABOCC) lasers.** The laser linewidth is in units of the generalized Schawlow–Townes linewidth. Engineering of the atom-cavity coupling and the photon loss allows both the SGBO and the ABOCC lasers to achieve a linewidth significantly narrower than the best conventional laser. The dots represent the numerically calculated linewidth ratio. The dashed blue line represents $D/D_{ST} = 1$, the expected linewidth for the conventional laser in the $\langle n \rangle \to \infty$ limit. The orange and green dashed lines are linear fits on the log-log scale of the linewidth ratio versus the mean photon number $\langle n \rangle$ of the SGBO and ABOCC laser, respectively. The SGBO laser $D/D_{ST} \sim \langle n \rangle^{-0.914}$ (orange) and ABOCC laser $D/D_{ST} \sim \langle n \rangle^{-1.098}$ (green). The black dashed line is a guide to the eye with $D/D_{ST} \sim \langle n \rangle^{-1}$.

**Suppressing the loss noise.** We start by extending Wiseman's strategy for decreasing the laser linewidth to make the SGBO laser. Following Wiseman, we replace the linear inductive coupling between the atom and the cavity by the bare operator coupling

$$H_{a\text{-}c}^{(SGBO)} = g_2(\hat{\sigma}^+\hat{e} + \hat{\sigma}^-\hat{e}^+), \tag{4}$$

where $\hat{e}$ and $\hat{e}^\dagger$ are defined in Eq. (1). We extend Wiseman's scheme by setting the cavity loss super-operator to be

$$\hat{\mathcal{L}}_{c\text{-}tl}^{(SGBO)} = \Gamma_e\mathcal{D}[\hat{e}] + \Gamma_c\mathcal{D}[\hat{c}], \tag{5}$$

where $\Gamma_e$ controls the rate of loss by the bare operators, while $\Gamma_c$ controls the rate of the conventional loss mechanism. This extension can be thought as a form of bath engineering. A small amount of conventional loss is essential for stabilizing the laser as, without it, neither the rate at which photons are pumped into the cavity, nor the rate at which photons leave the cavity depends on the number of photons in the cavity, and hence the laser becomes unstable.

We can achieve this tremendous reduction in linewidth as the bare operator couplings allow photons to enter and leave the laser cavity without inducing phase noise (and hence they do not directly contribute to the linewidth). On the other hand, the photon number operator is conjugate to the phase operator, and therefore in the presence of only bare operator couplings, the distribution of photon numbers in the cavity becomes infinitely broad. Adding conventional loss makes the photon number distribution have finite width, thereby stabilizing the laser at the cost of introducing phase noise. While both the conventional and the bare operator loss are contributing to the laser luminosity, only the conventional loss is contributing to the laser linewidth, and therefore the ratio of the SGBO laser linewidth $D_{SGBO}$ to the generalized ST limit can be approximated by

$$\frac{D_{SGBO}}{D_{ST}} \sim \frac{\langle n \rangle\Gamma_c}{2(\Gamma_e + \langle n \rangle\Gamma_c)}. \tag{6}$$

Here, the factor of 2 in the denominator accounts for the elimination of pump noise, similar to the laser proposed in ref. [2].

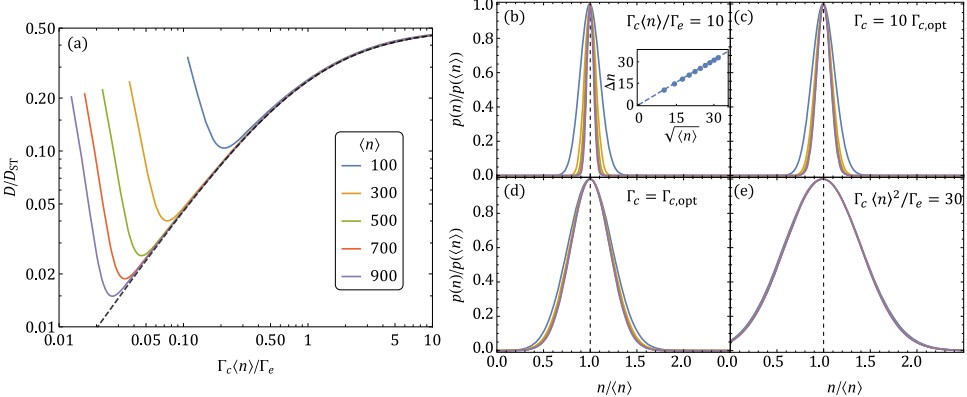

**Fig. 3 The linewidth and the photon distribution of the Susskind–Glogower Bare Operator (SGBO) laser. a** Minimum laser linewidth can be achieved by tuning the ratio of power emitted by conventional ($\langle n \rangle \Gamma_c$) and bare operator ($\Gamma_e$) loss for different cavity occupancies. The dotted line represents Eq. (6). Photon number distributions are shown in **b–e** for four cases: conventional loss is dominant (**b**); intermediate conventional loss (**c**); minimum linewidth (**d**); and conventional loss is very weak (**e**). The lines in all subfigures are for the SGBO laser with $\langle n \rangle = 100$ (blue), 300 (orange), 500 (green), 700 (salmon), and 900 (purple).

In Fig. 3a we plot the linewidth ratio $D_{\mathrm{SGBO}}/D_{\mathrm{ST}}$ as a function of $\Gamma_c \langle n \rangle / \Gamma_e$, the ratio between power emitted by conventional and bare operator loss while keeping the mean photon number fixed (see Methods). We observe that as we decrease $\Gamma_c \langle n \rangle / \Gamma_e$, the ratio $D_{\mathrm{SGBO}}/D_{\mathrm{ST}}$ first follows Eq. (6), then saturates at a point that depends on the number of photons in the cavity, and then begins increasing again. The origin of saturation and increase can be understood by looking at the distribution of photon numbers in the laser cavity.

When $\Gamma_c \langle n \rangle / \Gamma_e$ is large, conventional loss dominates and the distribution of photon numbers in the cavity has a width $\sim \sqrt{\langle n \rangle}$ (Fig. 3b). As $\Gamma_c \langle n \rangle / \Gamma_e$ decreases, the distribution of photon numbers in the cavity broadens (Fig. 3c). This continues until the distribution width $\Delta n$ becomes roughly half of $\langle n \rangle$, at which point the linewidth saturates and the photon distribution becomes universal (Fig. 3d). As $\Gamma_c \langle n \rangle / \Gamma_e$ is decreased even further, the photon number distribution becomes even broader (Fig. 3e) and the probability to have no photons in the cavity becomes appreciable. The state with no photons in the cavity does not have a well-defined phase. Consequently, the occupation of this state dominates the broadening of the laser linewidth for small $\Gamma_c \langle n \rangle / \Gamma_e$.

**Engineering the ABOCC coupling circuits**. We now take on the challenge of engineering the ABOCC laser using circuit QED devices. In the ABOCC laser system, we use a transmon qubit as a pumping atom (Fig. 1). We use bath engineering to achieve an incoherent drive on the transmon qubit to achieve population inversion (see Supplementary Note 1). The transmon qubit is modeled by a two-level system. The microwave cavity is modeled as a LC resonator, which couples to the transmon qubit and the output transmission line through two ABOCCs to mimic the SG operators.

We start with the atom-cavity coupling. The key property of the $\hat{e}$ operator is that the matrix element $|\langle n-1|\hat{e}|n\rangle| = 1$ is independent of $n$, while for the standard photon annihilation operator $\hat{c}$, $|\langle n-1|\hat{c}|n\rangle| = \sqrt{n}$. We have come up with the coupling circuit, depicted in Fig. 1, composed of an RF-SQUID (a Josephson junction shunted by a linear inductor) with an additional $\pi$ junction to ground (the $\pi$ junctions could be made from, for example, a second RF-SQUID that is flux-biased). The linear inductor in the RF-SQUID provides the linear coupling between the atom and the laser cavity (the cavity and the transmission line). The Josephson junction provides the non-linear coupling, whose strength is controlled by the Josephson

critical current of the junction. By tuning the critical current and the linear inductance, the Boson amplification factor ($\sqrt{n}$ factor) can be largely suppressed within some photon number range.

The ABOCC coupling the atom to the cavity is described by the Hamiltonian

$$H_{\mathrm{a\text{-}c}}^{(\mathrm{ABOCC})} = \frac{\phi_0^2 \hat{\delta}^2}{2 L_{\mathrm{a\text{-}c}}} - E_{\mathrm{J:a\text{-}c}} \cos \hat{\delta} + E_{\mathrm{J:c}} \cos \hat{\varphi}_c, \qquad (7)$$

where $\hat{\delta} = \hat{\varphi}_a - \hat{\varphi}_c$ and $\hat{\varphi}_a$ and $\hat{\varphi}_c$ are the superconducting phase operators of the transmon and the cavity; $\phi_0 = \Phi_0/2\pi$; $E_{\mathrm{J:a\text{-}c}} = \Phi_0 I_{\mathrm{J:a\text{-}c}}$ and $L_{\mathrm{a\text{-}c}}$ are the Josephson energy and the linear inductance of the RF-SQUID part of the ABOCC and $E_{\mathrm{J:c}}$ is the Josephson energy of the $\pi$ junction. After quantizing the microwave cavity field, applying Baker–Campbell–Hausdorff (BCH) formula and rotating wave approximation, the atom-cavity coupling induced by ABOCC is

$$H_{\mathrm{a\text{-}c}}^{(\mathrm{ABOCC})} = g\left(\hat{\sigma}^+ \hat{A}_{c,m_0} + \hat{\sigma}^- \hat{A}_{c,m_0}^\dagger\right), \qquad (8)$$

where $\hat{\sigma}$ are the Pauli operators for the atomic degree of freedom, the atom-cavity coupling strength $g$ is controlled by the critical current ($I_{\mathrm{J:a\text{-}c}}$) of the Josephson junction in the ABOCC and the normalization parameter $\mathcal{N}$ of the ABOCC cavity operator $\hat{A}_{c,m_0}$, $\hat{A}_{c,m_0}$ is an effective cavity photon annihilation operator,

$$\hat{A}_{c,m_0} = \frac{1}{\mathcal{N}}\left( \frac{2\phi_0^2 \tilde{\varphi}_a \tilde{\varphi}_c}{L_{\mathrm{a\text{-}c}} E_{\mathrm{J:a\text{-}c}}} c \right.$$
$$\left. - \sin(\tilde{\varphi}_a) e^{-\frac{\tilde{\varphi}_c^2}{2}} \sum_{n=0}^{\infty} \frac{(-1)^n \tilde{\varphi}_c^{2n+1}(c^\dagger)^n c^{n+1}}{n!(n+1)!} \right) \qquad (9)$$

and $\tilde{\varphi}_{a,c} = \frac{1}{\phi_0}\sqrt{\frac{\hbar Z_{\mathrm{a,c}}}{2}}$ and $\mathcal{N}$ is a normalization factor that ensures that $\langle m_0 | A_{c,m_0} | m_0 + 1 \rangle = 1$.

The operator $\hat{A}_{c,m_0}$ is designed to work when there are $m_0$ photons in the laser cavity, where $m_0$ depends on the transmon and cavity impedances (see inset of Fig. 4a and methods). In Fig. 4a we plot the matrix element $\langle n | \hat{A}_{c,m_0} | n+1 \rangle$ as a function of $n/m_0$. We observe that the matrix element has a plateau, centered on $n = m_0$, around which it is independent of $n$. The plateau is obtained by combining the sinusoidal current phase relation of the Josephson junction with the linear current phase of the inductor in the coupling circuit (see methods). On this plateau, $\hat{A}_{c,m_0}$ behaves approximately like the bare operator $\hat{e}$. In Fig. 4a the matrix element traces with different $m_0$s are

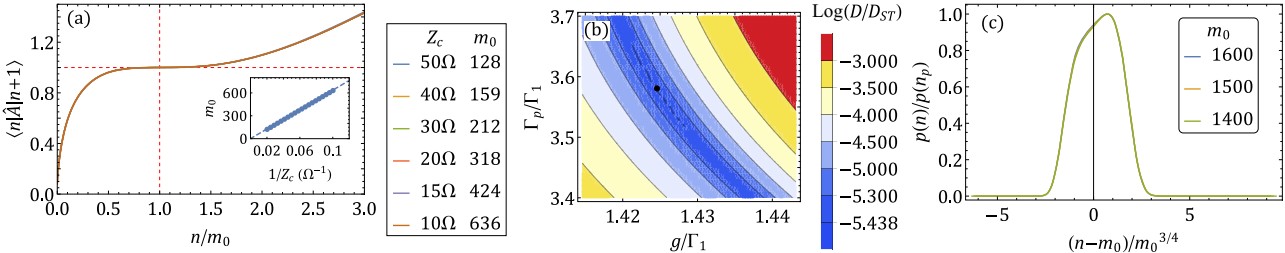

**Fig. 4 The Approximately Bare Operator Coupling Circuit (ABOCC) operator and the linewidth and photon distribution of the ABOCC laser. a** The matrix element $\langle n|\hat{A}_{c,m_0}|n+1\rangle$ as a function of the photon number in the cavity shows a plateau on which the matrix element is independent of the photon number thus approximating the bare operator. The matrix elements, which we computed for cavity impedance $Z_c = 50\,\Omega$ (blue), $40\,\Omega$ (orange), $30\,\Omega$ (green), $20\,\Omega$ (salmon), $15\,\Omega$ (purple), and $10\,\Omega$ (brown) which controls $m_0$ as shown in the inset (the impedance of the qubit was set to $Z_a = 47.71\,\Omega$), appear almost indistinguishable after re-scaling the photon number by $m_0$. **b** Tuning the ABOCC laser by varying the pump power $\Gamma_p$ and the atom-cavity coupling strength $g$ that is controlled by $l_{J:a\text{-}c}$. **c** The photon number distributions of the optimum linewidth point at $m_0 = 1600$ (blue), 1500 (orange), and 1400 (green). All three lines appear to collapse after appropriate re-scaling of both axes.

indistinguishable. This indicates that the range of $n$s over which $\hat{A}_{c,m_0}$ behaves like $\hat{e}$ also scales with $m_0$.

We introduce another ABOCC to engineer the nonlinear coupling between the laser cavity and the transmission line (bath). After applying BCH fomular and RWA, we further assume that the transmission line field can be treated as a Markovian bath and trace out the bath degrees of freedom (see Method). After tracing out the bath, the effective nonlinear loss for the cavity field can be expressed in the Lindblad form as follows:

$$\hat{\mathcal{L}}_{c\text{-}tl}^{\text{ABOCC}} = \Gamma_1 \mathcal{D}[\hat{B}_{c,m_0}], \tag{10}$$

where the rate constant

$$\Gamma_1 = \mathcal{N}^2 \frac{E_{J:c-tl}^2}{\hbar^2} \left(\frac{\hbar Z_{tl}}{2\phi_0^2}\right) \frac{1}{\omega_c} \tag{11}$$

the nonlinear cavity operator

$$\hat{B}_{c,m_0} = \frac{1}{\mathcal{N}} \left( \mathcal{C}_{tl} e^{-\tilde{\varphi}_c^2/2} \sum_{n=0}^{\infty} \frac{(-1)^n \tilde{\varphi}_c^{2n+1}}{n! \cdot (n+1)!} (c^\dagger)^n c^{n+1} \right.$$
$$\left. + \frac{2\phi_0^2 \tilde{\varphi}_c}{L_{c-tl} E_{J:c-tl}} c \right), \tag{12}$$

and $Z_{tl}$ is the transmission line characteristic impedance, $\omega_c$ is the frequency of the cavity, $\mathcal{C}_{tl} = \exp\left(-\frac{1}{\phi_0^2}\frac{\hbar Z_{tl}}{4\pi}\right)\left[1 + \frac{\theta}{\omega_c} + \frac{\theta^2}{2\omega_c^2} + O\left(\frac{\theta^3}{\omega_c^3}\right)\right]$ and $\theta \ll \omega_c$ is the bandwidth of the transmission line, $\mathcal{N}$ is a normalization constant (see Supplementary Note 4).

The ABOCC laser, in which we use ABOCCs for both atom-cavity and cavity-transmission line couplings, is stable and does not require conventional loss like the SGBO laser. To tune up the ABOCC laser we first choose the desired number of photons in the cavity and then set the impedances of the inverted transmon qubit, the cavity, and the transmission line so that both ABOCCs have the desired value of $m_0$. Next, we vary the transmon qubit's inverting incoherent pump strength $\Gamma_p$ (see[30,31] and Supplementary notes for how to implement this drive) and the atom-cavity coupling strength $g$ while fixing the cavity-transmission line coupling strength in order to minimize the ratio $D_{\text{ABOCC}}/D_{\text{ST}}$ (see Fig. 4b).

The general performance characteristics of the ABOCC laser are very similar to those of the SGBO laser. Both have a linewidth that is a factor of $1/\langle n\rangle$ narrower than the generalized ST limit (see Fig. 2). At the optimal operating point, both have a photon

distribution in the cavity with a width that scales with $\langle n\rangle^{3/4}$ as opposed to $\langle n\rangle^{1/2}$ for conventional lasers (see Figs. 3d and 4c and Supplementary Note 6). There is, however, a difference in the shape of the distributions. In the case of the SGBO laser, the photon distribution width is limited by occupation of the empty state. On the other hand, in the ABOCC laser, it is limited by the width of the plateau on which the operator $\hat{A}_{c,m_0}$ behaves like the bare operator (see Fig. 4a).

## Discussion

We pause to comment on what is a laser and what are the crucial ingredients for building a sub-SQL laser. We start by asking what are the key properties a laser: is it a lasing threshold? is it stimulated emission? or is it turning broad-band pump light into narrow linewidth output light? Both the device in ref.[23] and our device can be thought of as single-atom masers, in which just one atom as opposed to a collection of atoms is used to pump the resonant cavity. There is considerable literature on the properties of conventional single-atom lasers and masers and whether these devices should be considered to be true lasers though they do not have a lasing threshold[10,32–38]. However, we concentrate on single-atom lasers as a more technically tractable approach to the problem of constructing a sub-SQL device. Instead, the crucial point is abandoning both stimulated emission and conventional photon loss from the resonant cavity, which are the key features of both conventional and single-atom laser, in favor of quantum-engineered atom-cavity and cavity-output couplings. While the sub-SQL masers proposed here and in ref.[23] do not have a lasing threshold nor employ stimulated emission, they do share the crucial property that they can take noisy input light and turn it into ultranarrow output light as required by refs.[2,37].

Our results raise the question: given our demonstrated ability to greatly exceed the SQL, what is the ultimate quantum limit on the linewidth of a laser? Baker et al. argue that the ST limit divided by $\langle n\rangle^2$, which the authors call the Heisenberg limit, is the ultimate limit on laser linewidth[23,39]. Our laser circuit already far exceeds the ST limit, operating at the geometric mean of the ST and the Heisenberg limit. However, in passive nonlinear optical/microwave systems, there appears to still be some room for future improvements. Specifically, in contrast to ref.[23], where the photon number noise scales as $\langle n\rangle$, our proposed ABOCC laser shows $\langle n\rangle^{3/4}$ scaling. We speculate that by further optimizing the coupling circuits, it should be possible to realize photon number noise that scales with $\langle n\rangle$ and laser linewidth that approaches the Heisenberg limit.

In summary, we have shown that by engineering the photon loss operator, it is possible to build a laser that is $\langle n\rangle$ times

narrower than the SQL, where $\langle n \rangle$ is the number of photons in the laser cavity. We have also developed a realistic roadmap for constructing this type of laser using standard circuit QED components: capacitors, inductors, and Josephson junctions. These are exactly the same components that are used in a wide variety of superconducting quantum information devices. The device we propose could be an ultra-coherent, cryogenic light source for microwave quantum information experiments.

Furthermore, the photon field in the cavity of the proposed laser is highly squeezed (see Supplementary Note 6) with the photon number distribution width scaling with the photon number (as opposed to the square-root of the photon number that is observed in conventional lasers). The proposed device can be thought of as approaching the Heisenberg limit on phase estimation. We envision that the proposed device could be modified to provide designer quantum light that is an important resource for continuous variable/linear optical quantum computing[40–42], readout of quantum states in superconducting quantum computers[43,44], quantum metrology[45–48], and quantum communication[49–52]. In addition to being an interesting source of quantum light for quantum information experiments, the proposed device also shows how even well-understood quantum optical objects such as lasers can be re-imagined with the techniques of quantum information and the tools of superconducting circuits.

## Methods

**Eigenspectrum method to solve laser linewidth.** To describe the lasers we use the master equation

$$\dot{\rho} = \hat{\mathcal{L}}_a[\rho] - \frac{i}{\hbar}[H_{a\text{-}c}, \rho] - \frac{i}{\hbar}[H_c, \rho] + \hat{\mathcal{L}}_{c\text{-}tl}[\rho]. \tag{13}$$

The Hamiltonian $H_{a\text{-}c}$ describes that atom-cavity coupling and the super-operator $\hat{\mathcal{L}}_{c\text{-}tl}[\rho]$ describes loss of cavity photons to the transmission line. The super-operator $\hat{\mathcal{L}}_a[\rho] = -\frac{i\omega_a}{2}[\sigma^z, \rho] + \Gamma_p \mathcal{D}[\sigma^+]\rho$ describes the artificial atom, where $\sigma^z$ is the atom Pauli matrix and $\mathcal{D}[\sigma^+]\rho = \sigma^+\rho\sigma^- - \frac{1}{2}(\sigma^-\sigma^+\rho + \rho\sigma^-\sigma^+)$ describes the action of the incoherent pump. The incoherent drive on the transmon qubit is achieved through bath engineering. We couple a SNAIL qubit, which has third-order nonlinearity to achieve two-photon pumping process, to the transmon qubit. We drive the two-photon process that pumps both SNAIL and transmon qubits from their ground states to the first excited state. The SNAIL qubit also couples to a lossy cavity, which makes the SNAIL qubit to have a fast population decay back to the ground state. Therefore, the transmon qubit experiences an effective incoherent drive from its ground state to the excited state (see Supplementary Note 1). The Hamiltonian $H_c = \omega_c c^\dagger c$ describes the cavity energy levels, where $c$ is the photon annihilation operators in the cavity and we set $\omega_c = \omega_a$.

To obtain the linewidth of the laser we numerically find the eigenspectrum of the time evolution super-operator defined by the right hand side of Eq. (13). The master equation Eq. (13) can be thought of as an eigenvalue problem of the super-operator. The spectrum of eigenvalues has one zero eigenvalue $\lambda_0 = 0$, which corresponds to the steady-state solution of the laser system, and a number of negative eigenvalues that correspond to the decaying modes. The laser linewidth can be calculated by the Fourier spectrum of the two-time correlation function of the light field, which consists of a linear combination of decaying exponentials with the decay time set by these negative eigenvalues. Almost all of the weight is carried by the eigenstate with the largest nonzero eigenvalue of the super-operator (see Supplementary Note 3 for details). We use this largest nonzero eigenvalue of the super-operator to estimate the laser linewidth.

**Optimizing SGBO laser.** To optimize the SGBO laser (Fig. 3a): (1) we work in the strong atom-cavity coupling regime by fixing $g_2 = 1000\sqrt{\Gamma_p \Gamma_e}$. In this regime we estimate the mean photon number to be $\langle n \rangle \sim (\Gamma_p - 2\Gamma_e)/(2\Gamma_c)$. (2) We tune $\Gamma_p/\Gamma_e$ and $\Gamma_c/\Gamma_e$ to optimize the laser linewidth while fixing the mean number of photons in the cavity.

**Deriving ABOCC nonlinear cavity operators.** The ABOCC coupling the atom to the cavity is described by the Hamiltonian in Eq. (7). Note that the nonlinear atom-

cavity coupling is given by $\cos\hat{\delta}$ term, which can be expanded as follows:

$$\cos\hat{\delta} = \frac{1}{2}e^{-\frac{\tilde{\varphi}_c^2}{2}}\left(e^{-i\tilde{\varphi}_a\hat{\sigma}_x}e^{i\tilde{\varphi}_c c^\dagger}e^{i\tilde{\varphi}_c c} \right. \\ \left. + e^{i\tilde{\varphi}_a\hat{\sigma}_x}e^{-i\tilde{\varphi}_c c^\dagger}e^{-i\tilde{\varphi}_c c}\right) \tag{14}$$

where we assume the atom is a two-level system, whose operators can be expressed using Pauli operators, i.e., $\hat{\varphi}_a = \tilde{\varphi}_a\hat{\sigma}_x$, and $\tilde{\varphi}_{a,c} = \frac{1}{\phi_0}\sqrt{\frac{\hbar Z_{a,c}}{2}}$. This term induces a nonlinear self-energy of the cavity field $\cos(\tilde{\varphi}_a)\cos\hat{\varphi}_c$ that can be canceled by tuning the critical current of the $\pi$-junction in ABOCC. The rest of the terms with the cavity operators can be expanded using the BCH formula, and applying the rotating wave approximation we obtain Eq. (8).

The nonlinear coupling between the cavity and transmission line through ABOCC is analyzed using similar method. At first, the free field of the transmission line is quantized (see Supplementary Note 2 for details)[16]. The ABOCC coupling Hamiltonian is in the similar form of Eq. (7), except the phase difference operator

$$\hat{\delta}_{c\text{-}tl} = \tilde{\varphi}_c(\hat{c} + \hat{c}^\dagger) + i\sum_k \tilde{\varphi}_{tl}(k)\left(\hat{b}_k - \hat{b}_k^\dagger\right). \tag{15}$$

where $\tilde{\varphi}_{tl}(k) = \frac{1}{\phi_0}\sqrt{\frac{\hbar Z_{tl}}{2}}\sqrt{\frac{v_p}{l\omega_k}}$ is the transmission line quantization constant, $v_p$, $Z_{tl}$, and $l$ are the wave speed, the characteristic impedance, and total length of the transmission line, respectively.

Because of the nonlinear nature of the cavity-transmission line coupling, there are multi-photon exchange processes, which can be seen from the expanding the $\cos(\hat{\delta}_{c\text{-}tl})$ terms in the coupling Hamiltonian. We re-order the expanded terms by the number of transmission line (bath) operators, which is also the number of photon exchange between the cavity and the transmission line. We assume the transmission line is a vacuum bath and apply RWA and Markov approximation to trace out the transmission line degrees of freedom, all the terms in different orders in the coupling Hamiltonian give decay channels, which are represented by different Lindblad operators. The first-order terms in the Hamiltonian give the leading order among these decay channels (see Supplementary Note 4). We focus on the leading order and ignore the higher-order terms. The first-order term gives the Lindblad operator given in Eq. (10) with loss constant and nonlinear cavity operator in Eqs. (11) and (12).

## Data availability
The data used for generating all the figures in both the main manuscript and the supplementary notes are generated by the codes.

## Code availability
The code written in this study have been deposited in the Zenodo under https://doi.org/10.5281/zenodo.5016168[53].

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

## Acknowledgements

We thank Andrew Daley, John Jeffers, and Howard Wiseman for helpful comments. C.L. acknowledges support from a Pittsburgh Quantum Institute graduate student fellowship. Research was also supported by the Army Research Office under Grants Number W911NF-18-1-0144 and W911NF-15-1-0397, and by M. Hatridge's NSF CAREER grant (PHY-1847025). M.V.G.D. was partially supported by NSF EFRI ACQUIRE 1741656. The views and conclusions contained in this document are those of the authors and should not be interpreted as representing the official policies, either expressed or implied, of the Army Research Office or the U.S. Government. The U.S. Government is authorized to reproduce and distribute reprints for government purposes notwithstanding any copyright notation herein.

## Author contributions

C.L. derived the master equation, wrote and ran the numerical codes, analyzed the results, and produced the figures. M.V.G.D. pointed out Wiseman's work from 1999 that inspired this project. Following this inspiration, all co-authors contributed to the discussions that led to the development of the physical picture. M.M., X.C. and M.H. invented the two-qubit artificial atom and provided experimentally realizable parameters that were used by C.L. in maser calculations. D.P. and C.L. invented the ABOCC circuit and wrote the manuscript with input from all the co-authors. D.P. supervised the project.

## Competing interests

The authors declare no competing interests.
