## [Peer Review File · Nature Communications]

REVIEWER COMMENTS

Reviewer #1 (Remarks to the Author):

My previous comments have been adequately addressed, and I believe this is valuable work of broad interest that should be published. However, there are some (relatively minor) points that should be addressed before publication.

A point that the other referee brought up was that the authors said (in the abstract) that the scheme produces squeezed light. The authors have attempted to address this, but there seems to be some miscommunication. The way the authors have worded their abstract makes it sound like the OUTPUT of the laser is squeezed, and this is what the other referee assumed. What the authors say on page 5 is "The photon field in the cavity of the proposed laser is highly squeezed", and indeed that seems to be what the authors are analysing in the new appendix. Squeezing in the cavity and output field are quite distinct, and one does not imply the other. According to the analysis of Baker et al, one can expect that the cavity field is phase squeezed, because it needs to store phase information with a small number of photons. In contrast, the output field should have a much larger number of photons than the cavity state, but one should not be able to obtain a better phase estimate from the output than from the cavity field (because all phase information ultimately comes from the cavity). If the output were phase squeezed in the same way as the cavity field then it would mean there is more phase information in the output than the cavity, which is impossible.

All the authors need to do is make clear in the abstract that it is squeezing in the cavity. The new appendix does not seem to be needed, because squeezing in the cavity is exactly what would be expected and does not need to be strongly justified.

Other than that, I have some minor comments, mostly on the presentation.

1) Equation (2) is described as being obtained from the standard Schawlow-Townes formula by replacing the cavity linewidth with the ratio of luminosity to energy. There are a range of ways of giving the Schawlow-Townes formula, though, so the form they are deriving it from should also be given.

2) In figure 2 it should be explained what the dashed lines are representing.

3) Equation (5) is not adequately justified. For example, there is a factor of 2 there which is not mentioned, but is presumably from the gain similarly to [4]. It is also not made clear that this is an approximation.

4) The caption of figure 3 says the opposite of what would be expected. It looks like conventional loss is dominant in (b) and weak in (e), but the caption says that it is the other way around.

5) In figure 4 (a) and (c) the lines are indistinguishable, which makes it at first appear that there is a mistake and only one line has been plotted. It would be better to add a mention of this in the caption.

6) In the text it says

"We remark that the fact that in Fig. 4a the matrix element traces with different m_0 's collapse indicates that the range..."

This wording is ambiguous. A better wording would be

"In Fig. 4a the matrix element traces with different m_0 's are indistinguishable. This indicates that the range..."

7) Equation (A3a) is missing an equals sign.

8) In (C1) the absolute value of tau should be used on the right so that negative values of tau can be used in the integral in (C2).

9) Below (C2) the typo "Lorentzian" appears.

10) Equation (C9) has a superscript (i) in the first term, where it apparently should be (0).

11) Below equation (D19) there is $U=\exp(H_0)$, which is apparently missing an i in the exponential.

12) Equation (D30) is missing the index of summation q on the second sum.

13) The wording "descent suppression" does not make sense. Presumably the authors mean "decent". This wording appears near the bottom of page 22, near the top of page 23, and another two times on page 24. The wording "descent perturbation" also appears on page 24.

Reviewer #2 (Remarks to the Author):

I appreciate the detailed rebuttal letter describing both responses-in-principle to the referee comments, as well as actions taken to update the manuscript. Based on these and a reading of the new manuscript I recommend publication without further delay. This judgment is based mainly on my opinion that the proposed design of a sub-SQL circuit-QED laser that leverages stationary and potentially robust ABOCC couplers is a significant contribution to quantum engineering. I would note that there are some typos in the manuscript:

- 1) first paragraph after abstract - "Previous" capitalized unnecessarily
- 2) third paragraph of main text, "Baker et al" lacks a period after "al" although I guess this is a stylistic choice for the journal
- 3) same paragraph, "We engineer" is unnecessarily capitalized
- 4) first full paragraph after equation (1), the exclamation after "... narrower than the ST limit" seems gratuitous
- 5) caption of Fig. 2 should have an explanation of the dashed lines
- 6) first full paragraph on page 3, "All data in this figure was" should probably read "All data ... were", no?

****END****

RESPONSE TO REFEREES

We thank both the referees for the comments which have helped us to improve our manuscript. We address the comments and explain the corresponding modifications on our manuscript in detail below.

Referee 1: This is an interesting manuscript that proposes a method to provide laser linewidth beyond the Schawlow-Townes limit by using a modified coupling to the output using Josephson junctions. The Schawlow-Townes limit has been regarded as the fundamental limit for many years, so it is of great interest to be able to beat that limit. On this basis the work may be suitable for Nature Communications.

More specifically, the Schawlow-Townes limit is that the linewidth of a laser beam should scale no better than $1/n^2$, where n is the photon number inside the laser cavity. In this work the linewidth scales as $1/n^3$, potentially providing far greater coherence. This proposal is at microwave frequencies, so would correspond to a maser rather than a laser. The method corresponds to modifying the driving of the cavity, as well as the coupling to the output beam, in a way that corresponds to replacing the usual creation and annihilation operators with ones that have modified constant factors (in contrast to the usual square roots of factorials). For the driving that is equivalent to work by Wiseman [4], but the authors also modify the coupling to the output, which is what yields the improved scaling.

Something which is of interest is the relation to the recently released work by Baker et al. in Nature Physics. That work was published shortly after the submission of this manuscript. In that work a comparable proposal was given, but with linewidth scaling as $1/n^4$ instead of $1/n^3$ (i.e. better than the current manuscript). That work is based on a similar principle that the operators coupling the cavity to the output are modified from the standard annihilation and creation operators. A drawback was that the proposal is pulsed, rather than continuous wave, as is the proposal in this manuscript. Therefore this manuscript provides a more practical proposal than that of Baker et al., even if it does not achieve the same scaling.

It would be useful for the authors to compare their result with the proposal of Baker et al [T.J. Baker *et al*, Nat. Phys. (2020)] so that readers understand the relationship.

Response and Action: We agree with the referee. We have add a paragraph on Page 1 (the third paragraph of the main text after the abstract) of our manuscript to discuss Baker et al's most recent paper. We also compare the Baker et al's paper and our main result to reflect the differences, especially in the experimental proposals.

Referee 1: Another issue is that the title of this manuscript may cause confusion, because it is "A New Quantum Limit on Laser Linewidth", whereas the title of the Baker et al. paper is "The Heisenberg limit for laser coherence". (The linewidth and coherence are inversely proportional.) That makes it sound like these two works have the same results, but this manuscript does not actually prove a limit. In contrast, the Baker et al. paper is primarily on proving a limit, and the proposal for how to achieve that scaling is a more minor part of that work. This means that these two works are not as similar as is suggested by the title, and changing the title could help to make the difference clearer.

Response and Action: In light of the publication of Baker et al manuscript, we agree with the referee's point about the title of our manuscript. See also the answer below. We have therefore revised the title to a "Proposal for a continuous wave laser with linewidth well below the standard quantum limit".

Referee 1: Something that seems odd in this manuscript is that near the end it refers to earlier talks on this topic by the authors of the Baker et al. paper, saying "During the final stages of preparation of our manuscript we became aware of a pair of recent talks". That is normally done when referring to work that was released almost concurrently, but the earlier talk is from a conference that was held 4-8 August 2019, which predates submission of this work by more than a year. The cited talk has a 3-page extended abstract. A further search brings up another talk from 2019 that is uncited. That is at the conference AQIS, which was held 19-23 August 2019, and there is a book of abstracts at <http://newton.kias.re.kr/~jaewan/AQIS2019/t.pdf> There is another 3-page abstract in that book. It would normally be expected that prior work is mentioned and cited at the beginning in order to give the new results in appropriate context.

Note that those abstracts in 2019 explained how the limit could in principle be achieved, but did not provide an experimental proposal. The more recent talk cited is from 28 July 2020. There is no printed abstract apparent, but there is publicly available video more than an hour long. In that video the experimental proposal that later appeared in the Baker et al. paper was explained. The two proposals are explained in very different ways, so appear to have been arrived at independently.

Response and Action: While we agree with the spirit of the referee’s comment we want to make several points. (1) At the time when we submitted our manuscript for publication, Baker et al manuscript was not yet public (it was not posted on the arxiv nor available in Nature Physics). (2) We were aware of the two talks but not the 3-page abstract in the book. The video for the most recent talk was broken so while we knew about its existence we did not appreciate its content. Now that we have seen the Nature Physics paper, we agree with the referee that it deserves to be discussed on Page 1, and we have moved the discussion there. Because we think the main content in the previous talks are covered by the most recent paper, we only focus on the Nature Physics paper instead (this change is consistent with our communications with H. Wiseman). Further, we slightly modify the discussion at the end of the manuscript (on page 5, the second paragraph from the last) to reflect the changes.

Referee 1: Another issue with this work is that the Supplementary Information seems somewhat disorganised. I was expecting to see derivations of the equations given in the Methods, but these appear in different forms with different variable names. For example, Eqs. (10) and (11) are similar to (D12) and (D10) in the Supplementary Information, but have unexplained differences. Another difference is that the phi with subscript a is used for the transmon in Eq. (9). In the Supplementary Information, there is a phi with a subscript T used instead, which has a different definition. The Supplementary Information needs to be made consistent with the main text, and refer back to equations in the main text that are being derived.

Response and Action: Throughout the Appendix B and Appendix D, we change the subscript ‘T’ to ‘tl’ to reflect the corresponding quantities are for the transmission line, which is consistent with the main text. In addition, we also relabel some Josephson energy constant E_J with appropriate corresponding subscript, e.g., $E_{J:a-c}$, $E_{J:c-tl}$ for atom-cavity and cavity-transmission line coupling Josephson junction energy.

To address the inconsistency of the notations between the (old) Eqs. (10) and (11) and the appendix Eq. (D12) and (D10), on Page 21, Appendix D, at very last of the section, we add a new paragraph to highlight the link between the derived equations in this appendix section to the equation for the nonlinear loss of the cavity field we used in the main text.

As we are discussing the ABOCC coupling between the cavity and transmission line in our Appendix D, to be consistent with the main text, we change all the nonlinear operators for the cavity field from \hat{A} to \hat{B} .

Referee 1: Some other issues in this work,

(1) In the Supplementary Information, just above Eq. (B5) it says “The coupling Hamiltonian is given in Eq. (7) of the main text”, but that is not a coupling Hamiltonian. There is a coupling Hamiltonian in Eq. (6), but it does not explicitly have the parameters described in Eq. (B5). Those are not introduced until Eq. (9) on the next page.

Response and Action: On page 11, Appendix B, after Eq. (B4), we notice that the original reference to the main text Eq. (7) is incorrect. In section Appendix B, the discussion is about the linear coupling between the cavity and the transmission line, which does not corresponding to any equations in the main text. We added a new equation for the Hamiltonian of the linear inductive cavity-bath coupling to the appendix, which is the new Eq. (B5).

Referee 1: (2) Just above Eq. (10), B has a subscript n_p , whereas elsewhere it is m_0 .

Action: We change the subscript of \hat{B} to m_0 .

Referee 1: (3) Just above Eq. (D2) it says “see Section B of the online supplement”, which is odd if this document is intended to be the online supplement. It would make more sense to just say “see Section B”.

Action: We change it to “Appendix B”.

Referee 1: (4) There are some obvious spelling mistakes. In reference [41], “Heisenberg” is spelt “Heisenburg”, in the caption of figure 5 it says that the levels are “weekly populated”, and again in that caption it says “think solid lines”.

Action: We fixed these errors.

Referee 2: The current manuscript presents several notable results; among them, the authors choose in their presentation to emphasize their conclusion that realizing SG-type operators for both the atom-cavity and cavity-TL couplings leads to a laser linewidth exceeding the ST limit by $\langle n \rangle^2$. Additional notable results are presented mainly in the supplementary material, related to analysis of the behavior of their ABOCC construction.

In my opinion the manuscript should be accepted as these results are of significant interest for quantum engineering,

but the authors should consider a shift of emphasis in their presentation. I haven't really followed the literature on "standard quantum limits" for laser linewidth but while it is certainly easy to appreciate the potential significance of the result that adding SG cavity-TL couplings to the Wiseman scheme further lowers the emission linewidth, I think that a manuscript focused on this apparent fact should read rather differently – one would expect a more detailed analysis of why/how the change in output coupling achieves this effect, as well as a much more detailed consideration of prior literature on sub-SQL lasers (including for example single-atom and single-artificial-atom cavity/circuit-QED "lasers" and associated debates on what sorts of devices should and shouldn't be called "lasers").

Response and Action: We thank the referee for his/her comment that our discussion of laser linewidth should include more details about the previous literature and how to define what is and is not a laser. To address these points and support our follow-on discussion on the proposed laser system, we added a new paragraph (last paragraph on page 1), in which we comment on the definition of a laser and review previous progresses on single-atom lasers. Also, as requested by the Referee 1, we added a new paragraph (paragraph 3 on Page 1) to discuss a recent paper about the new limit on the laser linewidth from Baker et al (see above).

Referee 2: In my view, I would consider the detailed analysis of the ABOCC construction to be a more significant, and more broadly-interesting accomplishment than simply integrating master equations with SG coupling operators. I would suggest that the authors promote some of this material from the Supplementary to the main paper, including in particular more detail on the derivation of equation (9) as well as the more nuanced analysis of the laser linewidth (replacing the brief statement in the current "Methods" section that the largest negative eigenvalue of the Liouvillian superoperator gives the laser linewidth, which the authors themselves show is only true in an approximate sense). I think the demonstration that nonlinear circuit elements can be used in a microwave (or potentially nanophotonic) setting to engineer novel output coupling behaviors of nonlinear devices is actually of broader significance than the results of Fig 3a, and in that sense the material currently contained in the authors' Supplementary is in some sense of greater interest.

Response: We thank the referee pointing out that the quantum engineering we used to build the system could also be widely interesting to a broad audience. To stress the quantum engineering we used in the ABOCC laser systems, we added the key details to the main text of the manuscript. Limited by the length of the article, we were not able to fit an exhaustive discussion of these topics in the main text and hence this level of detail remains in the appendices.

Action: We made the following modifications of our main manuscript.

1. We explicitly state that the incoherent drive on the transmon qubit is achieved using bath engineering in the second paragraph from the last on Page 3. The bath engineering on the transmon qubit pumping scheme is also briefly sketched in the Method section (the first paragraph in Method section on page 6).
2. We explicitly demonstrate the essence of the ABOCC coupling circuit design in the last paragraph and the one paragraph above on Page 3.
3. We give the exact Hamiltonian for the ABOCC coupling as new Eq. (6) on page 3 and discuss briefly how we get the coupling Hamiltonian with nonlinear cavity operator, as well as the expression for the nonlinear cavity operator \hat{A}_{c,m_0} as new Eq. (8). A details of how we expand the cosine of the phase operator is added as the fourth paragraph in the Method section.
4. We show the exact expression for the nonlinear cavity decay rate constant Γ_1 and the corresponding nonlinear cavity operator \hat{B}_{c,m_0} as new Eqs. (10) and (11). We added a more detailed discussion to the Methods section as the last two paragraphs.
5. We have rewritten the paragraph in the Methods section (the second paragraph) about the laser linewidth calculation to sketch the method with more details.

Referee 2: As a minor additional comment, I would note that the reference chosen in the introduction for optical cavity QED is a bit odd as that review says in its abstract that it focuses mainly on the microwave maser. Likewise, the text around it states that strong coupling is achieved "at optical frequencies" by extending the time for light-matter interaction, but I think it's fair to say that in practice the more important mechanism for strong coupling in optical cavity QED has been reduction of the cavity mode volume (increasing single-photon Rabi frequency).

Response and Action: We change the citations in the discussion of optical cavity QED in the introduction to match the content we are discussing.

For the strong coupling regime in the cavity QED systems, we meant to state that the effective coupling time, which is the inverse of the Rabi frequency of the coupling, is long compared to the decay time of the cavity. This is essentially the same as atom-cavity coupling strength (characterized by Rabi frequency) is stronger than the cavity decay (high Q cavity). We believe this is essentially the same as the referee's point. To avoid further confusion caused by the language in our manuscript, we modified it as the referee suggested.

Referee 2: Continuing with minor concerns about the abstract/introduction, there seems to be some implication in the abstract/introduction that achieving sub-SQL microwave laser linewidths would be technologically significant, but there is no mention in the manuscript regarding the robustness of this scheme to technical noise/fab-error/drifts or of what sorts of actual Allan variances might be achievable. How would a scheme like this compare to state-of-the-art alternatives?

Response: We added a new section in the supplementary material, Appendix E, with a discussion of the robustness of our proposed ABOCC laser system to imperfections. We show that the narrow laser linewidth of our proposed ABOCC laser has some tolerance to errors in the atom-cavity coupling strength, pump strength on the transmon qubit and inductance ratio inside the ABOCC operators. Obtaining the narrowest possible linewidth in the ABOCC laser does require the transmon qubit frequency to match the laser cavity frequency. We suggest that this can be achieved by using a frequency-tunable transmon qubit as the pump atom, so that an external magnetic flux can be used to fine tune the transmon qubit frequency to match the laser cavity frequency. With these notions in mind, we think that the proposed ABOCC laser system can be fabricate in the lab. A more detailed discussion of device-level simulation depends on detailed design choices. These choices include the properties of qubits (frequency, lifetime, and non-linearity), type of resonant cavity (3D cavity, co-planer, etc), and the implementation of coupling between the elements (inductive vs. capacitive). We think that these design choices are beyond the scope of the present manuscript.

Further, the ABOCC laser proposal explored in our paper is not optimized to achieve narrowest possible laser linewidth. The details of what can be achieved are strongly dependent on experimental constraints like the number of photons in the lasing cavity and the desired laser luminosity. However, the main point demonstrated in our manuscript is that it should be possible to construct lasers with linewidth well beyond the ST limit. Therefore, we did not include a comparison between our ABOCC laser and the state-of-the-art laser linewidths.

Action: We added a new section in Appendix (Appendix E) to discuss the effect of the possible errors on the ABOCC laser system due to the possible experimental and device fabrication errors. In Appendix E.1, we discuss the possible error on the coupling strengths and pump strength. In Appendix E.2, we discuss the error on the inductance ratio in ABOCC operators. In Appendix E.3, we consider the possible fabrication errors which causes the transmon qubit is detuned from the cavity frequency.

Referee 2: The abstract mentions that the present scheme generates “highly squeezed” output fields but I don't think the main text analyzes this sufficiently.

Response: We have added a new section to the supplement, Appendix F, in which we analyze the properties of the effective photon lowering operators. In contrast to the conventional photon lowering operator, that has coherent eigenstates, we show that our effective photon lowering operator \hat{A} has phase squeezed eigenstates.

Action: We add a new section in Appendix as Appendix. F to discuss the characters of the eigenstates of the ABOCC operator. We plot the Wigner distribution of the eigenstates of different ABOCC operators, especially the eigenstates that best-match the photon distribution of the laser states that have best linewidth performance. The Wigner distribution of the ABOCC operator eigenstates shows the squeezing along the phase direction. We also calculate the phase noise of the ABOCC lowering operator as we increase the m_0 value (controlled by the cavity impedance Z_c) of the ABOCC operator.

While addressing the referee's question, we noticed there was a mistake in our main manuscript regarding the scaling of the photon noise. We incorrectly stated that Δn scales with n for the ABOCC laser. The correct scaling is $\Delta n \sim n^{3/4}$.

Action: We changed the photon number noise scaling in our ABOCC laser system from $\langle n \rangle$ to $\langle n \rangle^{3/4}$ on page 5, second paragraph, right column. We also changed Fig. 4c in the text to collapse the photon distribution using the updated scaling. A more detailed discussion is given in new Appendix F. We also added a comment on the photon number noise scaling in our proposal in comparison with Baker et al. (3rd paragraph on the right column on page 5).

REVIEWER COMMENTS

Reviewer #1 (Remarks to the Author):

My previous comments have been adequately addressed, and I believe this is valuable work of broad interest that should be published. However, there are some (relatively minor) points that should be addressed before publication.

A point that the other referee brought up was that the authors said (in the abstract) that the scheme produces squeezed light. The authors have attempted to address this, but there seems to be some miscommunication. The way the authors have worded their abstract makes it sound like the OUTPUT of the laser is squeezed, and this is what the other referee assumed. What the authors say on page 5 is "The photon field in the cavity of the proposed laser is highly squeezed", and indeed that seems to be what the authors are analysing in the new appendix. Squeezing in the cavity and output field are quite distinct, and one does not imply the other. According to the analysis of Baker et al, one can expect that the cavity field is phase squeezed, because it needs to store phase information with a small number of photons. In contrast, the output field should have a much larger number of photons than the cavity state, but one should not be able to obtain a better phase estimate from the output than from the cavity field (because all phase information ultimately comes from the cavity). If the output were phase squeezed in the same way as the cavity field then it would mean there is more phase information in the output than the cavity, which is impossible.

All the authors need to do is make clear in the abstract that it is squeezing in the cavity. The new appendix does not seem to be needed, because squeezing in the cavity is exactly what would be expected and does not need to be strongly justified.

Other than that, I have some minor comments, mostly on the presentation.

1) Equation (2) is described as being obtained from the standard Schawlow-Townes formula by replacing the cavity linewidth with the ratio of luminosity to energy. There are a range of ways of giving the Schawlow-Townes formula, though, so the form they are deriving it from should also be given.

2) In figure 2 it should be explained what the dashed lines are representing.

3) Equation (5) is not adequately justified. For example, there is a factor of 2 there which is not mentioned, but is presumably from the gain similarly to [4]. It is also not made clear that this is an approximation.

4) The caption of figure 3 says the opposite of what would be expected. It looks like conventional loss is dominant in (b) and weak in (e), but the caption says that it is the other way around.

5) In figure 4 (a) and (c) the lines are indistinguishable, which makes it at first appear that there is a mistake and only one line has been plotted. It would be better to add a mention of this in the caption.

6) In the text it says

"We remark that the fact that in Fig. 4a the matrix element traces with different m_0 's collapse indicates that the range..."

This wording is ambiguous. A better wording would be

"In Fig. 4a the matrix element traces with different m_0 's are indistinguishable. This indicates that the range..."

7) Equation (A3a) is missing an equals sign.

8) In (C1) the absolute value of tau should be used on the right so that negative values of tau can be used in the integral in (C2).

9) Below (C2) the typo "Lorentzian" appears.

10) Equation (C9) has a superscript (i) in the first term, where it apparently should be (0).

11) Below equation (D19) there is $U=\exp(H_0)$, which is apparently missing an i in the exponential.

12) Equation (D30) is missing the index of summation q on the second sum.

13) The wording "descent suppression" does not make sense. Presumably the authors mean "decent". This wording appears near the bottom of page 22, near the top of page 23, and another two times on page 24. The wording "descent perturbation" also appears on page 24.

Reviewer #2 (Remarks to the Author):

I appreciate the detailed rebuttal letter describing both responses-in-principle to the referee comments, as well as actions taken to update the manuscript. Based on these and a reading of the new manuscript I recommend publication without further delay. This judgment is based mainly on my opinion that the proposed design of a sub-SQL circuit-QED laser that leverages stationary and potentially robust ABOCC couplers is a significant contribution to quantum engineering. I would note that there are some typos in the manuscript:

- 1) first paragraph after abstract - "Previous" capitalized unnecessarily
- 2) third paragraph of main text, "Baker et al" lacks a period after "al" although I guess this is a stylistic choice for the journal
- 3) same paragraph, "We engineer" is unnecessarily capitalized
- 4) first full paragraph after equation (1), the exclamation after "... narrower than the ST limit" seems gratuitous
- 5) caption of Fig. 2 should have an explanation of the dashed lines
- 6) first full paragraph on page 3, "All data in this figure was" should probably read "All data ... were", no?

****END****

RESPONSE TO REFEREES

We thank both the referees for the comments which have helped us to improve our manuscript. We address the comments and explain the corresponding modifications on our manuscript in detail below. As part of our submission, we have included a version of our manuscript in which we highlight the changes in red to make it easier to keep track of them.

Referee 1: A point that the other referee brought up was that the authors said (in the abstract) that the scheme produces squeezed light. The authors have attempted to address this, but there seems to be some miscommunication. The way the authors have worded their abstract makes it sound like the OUTPUT of the laser is squeezed, and this is what the other referee assumed. What the authors say on page 5 is "The photon field in the cavity of the proposed laser is highly squeezed", and indeed that seems to be what the authors are analysing in the new appendix. Squeezing in the cavity and output field are quite distinct, and one does not imply the other. According to the analysis of Baker et al, one can expect that the cavity field is phase squeezed, because it needs to store phase information with a small number of photons. In contrast, the output field should have a much larger number of photons than the cavity state, but one should not be able to obtain a better phase estimate from the output than from the cavity field (because all phase information ultimately comes from the cavity). If the output were phase squeezed in the same way as the cavity field then it would mean there is more phase information in the output than the cavity, which is impossible.

All the authors need to do is make clear in the abstract that it is squeezing in the cavity. The new appendix does not seem to be needed, because squeezing in the cavity is exactly what would be expected and does not need to be strongly justified.

Response and Action: We thank the referee for pointing this out. We modified our abstract to stress that the light field inside the cavity of the proposed laser is highly squeezed in our abstract.

Referee 1: Equation (2) is described as being obtained from the standard Schawlow-Townes formula by replacing the cavity linewidth with the ratio of luminosity to energy. There are a range of ways of giving the Schawlow-Townes formula, though, so the form they are deriving it from should also be given.

Response and Action: We include the definition of the Schawlow-Townes limit as the new Eq. (2).

Referee 1: In figure 2 it should be explained what the dashed lines are representing.

Response and Action: We added further discussion about the dashed lines in our Fig. 2 (page 3) in the caption.

Referee 1: Equation (5) is not adequately justified. For example, there is a factor of 2 there which is not mentioned, but is presumably from the gain similarly to [4]. It is also not made clear that this is an approximation.

Response and Action: We have added a remark to make it clear that the equation is approximate and also added a sentence to explain the factor of 2 after Eq. (5) (new Eq. (6) in our manuscript) at page 3.

Referee 1: The caption of figure 3 says the opposite of what would be expected. It looks like conventional loss is dominant in (b) and weak in (e), but the caption says that it is the other way around.

Response and Action: We apologize that we forgot to update the caption after we re-grouped the sub-figures. In order to be consistent, we changed the caption of Fig. 3. We also modified the sentence to explicitly state that the relation is an approximation.

Referee 1: In figure 4 (a) and (c) the lines are indistinguishable, which makes it at first appear that there is a mistake and only one line has been plotted. It would be better to add a mention of this in the caption.

Response and Action: We thank the referee for pointing this issue out. We now mention that in the figure caption.

Referee 1: In the text it says "We remark that the fact that in Fig. 4a the matrix element traces with different m_0 's collapse indicates that the range..."

This wording is ambiguous. A better wording would be

"In Fig. 4a the matrix element traces with different m_0 's are indistinguishable. This indicates that the range..."

Action: We thank the referee for pointing it out. We modified our text as the referee suggested.

Referee 1:

- Equation (A3a) is missing an equals sign.
- In (C1) the absolute value of tau should be used on the right so that negative values of tau can be used in the integral in (C2).
- Below (C2) the typo “Lorentizian” appears.
- Equation (C9) has a superscript (i) in the first term, where it apparently should be (0).
- Below equation (D19) there is $U=\exp(H0)$, which is apparently missing an i in the exponential.
- Equation (D30) is missing the index of summation q on the second sum.
- The wording “descent suppression” does not make sense. Presumably the authors mean “decent”. This wording appears near the bottom of page 22, near the top of page 23, and another two times on page 24. The wording “descent perturbation” also appears on page 24.

Action: We thank the referee for pointing out the typos, which we have now fixed.

Referee 2: I appreciate the detailed rebuttal letter describing both responses-in-principle to the referee comments, as well as actions taken to update the manuscript. Based on these and a reading of the new manuscript I recommend publication without further delay. This judgment is based mainly on my opinion that the proposed design of a sub-SQL circuit-QED laser that leverages stationary and potentially robust ABOCC couplers is a significant contribution to quantum engineering. I would note that there are some typos in the manuscript:

1. First paragraph after abstract - ”Previous” capitalized unnecessarily
2. third paragraph of main text, “Baker et al” lacks a period after “al” although I guess this is a stylistic choice for the journal
3. same paragraph, “We engineer” is unnecessarily capitalized
4. first full paragraph after equation (1), the exclamation after “... narrower than the ST limit” seems gratuitous
5. First full paragraph on page 3, ”All data in this figure was” should probably read ”All data ... were”, no?

Response and Action: We thank the referee for the kind response and for pointing out these typos in our manuscript. We have corrected all of them.

Referee 2: Caption of Fig. 2 should have an explanation of the dashed lines.

Response and Action: This issue was also pointed out by the first referee. We have updated the caption of the Fig. 2 as suggested by the referees.